# Linear-Time Probabilistic Solutions
# of Boundary Value Problems

**Nicholas Krämer**
University of Tübingen
Tübingen, Germany
nicholas.kraemer@uni-tuebingen.de

**Philipp Hennig**
University of Tübingen and
Max Planck Institute for Intelligent Systems
Tübingen, Germany
philipp.hennig@uni-tuebingen.de

## Abstract

We propose a fast algorithm for the probabilistic solution of boundary value problems (BVPs), which are ordinary differential equations subject to boundary conditions. In contrast to previous work, we introduce a Gauss–Markov prior and tailor it specifically to BVPs, which allows computing a posterior distribution over the solution in linear time, at a quality and cost comparable to that of well-established, non-probabilistic methods. Our model further delivers uncertainty quantification, mesh refinement, and hyperparameter adaptation. We demonstrate how these practical considerations positively impact the efficiency of the scheme. Altogether, this results in a practically usable probabilistic BVP solver that is (in contrast to non-probabilistic algorithms) natively compatible with other parts of the statistical modelling tool-chain.

## 1 Boundary value problems in computational pipelines

This work develops a class of algorithms for solving *ODE boundary value problems*; that is, ordinary differential equations (ODEs)

$$\dot{y}(t) = f(y(t), t) \tag{1}$$

subject to *left-* and *right-hand side boundary conditions* $Ly(t_0) = y_0$ and $Ry(t_{\max}) = y_{\max}$. The vector field $f : \mathbb{R}^d \to \mathbb{R}^d$, as well as $L \in \mathbb{R}^{d_L \times d}, R \in \mathbb{R}^{d_R \times d}, t_0 \in \mathbb{R}, t_{\max} \in \mathbb{R}, y_0 \in \mathbb{R}^{d_L}$, and $y_{\max} \in \mathbb{R}^{d_R}$ are given. It is no loss of generality to consider a first-order boundary value problem because higher-order problems can be transformed into first-order problems [1].

Loosely speaking, solving BVPs amounts to following the law of a dynamical system when "connecting two points". This setting is relevant to several scientific applications of machine learning. As motivation, we consider three examples, all of which are depicted in Figure 1. First, recovering the trajectory of a pendulum between two positions amounts to solving the ODE $\ddot{y}(t) = -9.81 \sin(y(t))$ subject to the positions as boundary conditions. If the positions were interpolated without the ODE knowledge, the output would be physically meaningless. Second, BVPs arise when inferring the evolution of the case counts of people that fall victim to an infectious disease. A lack of counts of (a specific subset of) non-infected people at the initial time-point can be made up for by available counts of infected people at the final time-point of the integration domain. Third, efficient manifold learning necessitates repeated computation of (geodesic) distances between two points, which amounts to solving BVPs [2, 3]. Depending on application details, the ability to produce structured output uncertainty or to enhance the algorithm by including additional sources of information can be crucial. Probabilistic numerical algorithms respond to these challenges by solving problems of numerical simulation with probabilistic inference. For *initial* value problems, probabilistic solvers share linear-time complexity, adaptive step-size selection, and high polynomial convergence rates with their

35th Conference on Neural Information Processing Systems (NeurIPS 2021).

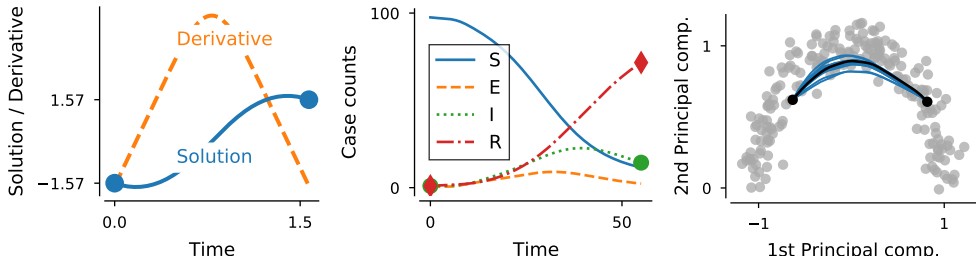

Figure 1: Recovering the trajectory of a pendulum between two positions is a BVP (LEFT). Lack of initial values can be made up by boundary values in an SEIR model (MIDDLE). Straight lines on manifolds give distance measures and demand solving a BVP (RIGHT; depicted are the mean and ten samples of the probabilistic solution; principal components of 1000 MNIST images of the digit "1"). In all three figures, the ball/diamond markers express boundary conditions.

non-probabilistic counterparts [4–7], and further provide functionality to quantify uncertainty within probabilistic programs [8, 9].

Probabilistic BVP solvers have not yet reached this level of quality. Existing probabilistic treatments of BVPs [10–12] iteratively condition a Gaussian process on approximately "solving the BVP". Each such iteration requires solving a generic least-squares problem of size equal to the number of employed grid points. The resulting cubic complexity puts severe upper limits on grid resolution. Traditional, non-probabilistic BVP solvers (for instance, those presented in [1]) are very efficient but do not provide probabilistic output. Thereby, they would have to serve as black-boxes inside probabilistic programs. In this work, we close this gap. The main idea of this paper is that computing a probabilistic solution of BVPs is fast if the prior is Markovian (Section 2). Probabilistic modelling provides additional advantages. In particular, algorithmic parameters can be estimated automatically (including those that must be provided by the user in traditional methods; Section 3–5).

## 2 Boundary value problems as probabilistic inference tasks

### 2.1 Generative model

Let $\sigma > 0$. We define the process $Y = [Y_0, ..., Y_\nu]^\top : [t_0, \infty) \to \mathbb{R}^{d(\nu+1)}$ as the solution of the stochastic differential equation

$$\mathrm{d}Y(t) = AY(t)\,\mathrm{d}t + B\,\mathrm{d}W(t), \quad Y(t_0) \sim \mathcal{N}(m_0, \sigma^2 C_0), \tag{2}$$

driven by a $d$-dimensional Wiener process $W : [t_0, \infty) \to \mathbb{R}^d$ with diffusion $\Gamma = \sigma^2 I \in \mathbb{R}^{d \times d}$, and initial parameters $m_0 \in \mathbb{R}^{d(\nu+1)}$, $C_0 \in \mathbb{R}^{d(\nu+1) \times d(\nu+1)}$ [4]. For the moment, we set $\sigma = 1$, $m = (0, ..., 0)$, and $C_0 = I$, and will discuss parameters calibration later. Let $A \in \mathbb{R}^{d(\nu+1) \times d(\nu+1)}$ and $B \in \mathbb{R}^{d(\nu+1) \times d}$ be such that the zeroth component $Y_0$ of $Y$ is the integrated Wiener process. The $q$th component $Y_q$ of $Y$ is the $q$th derivative of the integrated Wiener process. In this setup, $Y_0$ models the BVP solution $y$, and $Y_q$ models the $q$th derivative of the BVP solution $y$: $Y_q(t) \approx y^{(q)}(t)$, $q = 0, ..., \nu$ [6, Equation 2]. This is the prior for the probabilistic BVP solver. Other choices are possible, too [5, Section 2.1].

For ODE solvers, the likelihood is best described in terms of an information operator [13]. For BVPs, there are two sources of information: first, the *boundary conditions*

$$\ell_L(Y) := LY_0(t_0) - y_0 \quad \text{and} \quad \ell_R(Y) := RY_0(t_{\max}) - y_{\max}, \tag{3}$$

and second, the *differential equation*, encoded by the information operator

$$\ell(Y)(t) := Y_1(t) - f(Y_0(t), t). \tag{4}$$

Similar models are used in the gradient matching literature [14, 15]. Different to the likelihoods from conventional Bayesian inference, information operators used in probabilistic numerics are noise-free and often map between (possibly infinite-dimensional) vector spaces of functions [16, Section 2].

Let $\mathbb{T} := (t_0, ..., t_N = t_{\max})$ be a grid on $[t_0, t_{\max}]$. For now, we assume this grid is given; Section 4 introduces a strategy for iterative mesh-refinement based on error-control. We will abbreviate $\ell_n(Y) := \ell(Y)(t_n)$ and $\ell_{0:n} = (\ell_0, ..., \ell_n)$, $n = 0, ..., N$. Using $\mathbb{T}$, as well as the likelihoods in Equations (3) and (4), the approximate ODE solution is captured by the posterior distribution

$$p\left(Y(t) \,|\, \ell_L(Y) = 0, \ \ell_{0:N}(Y) = 0, \ \ell_R(Y) = 0\right). \tag{5}$$

Unfortunately, the full posterior (5) is intractable because of the non-linearity of $f$ (which implies non-linearity in all $\ell_n$). We will thus approximate it with a Gaussian: the *probabilistic BVP solution*.

## 2.2 Approximate Gaussian posterior inference

While the full posterior in Equation (5) cannot be computed in closed form, a maximum-a-posteriori (MAP) estimate is obtained by finding the minimum

$$\arg\min_{Y(\mathbb{T})} \left\{ -\log p(Y(\mathbb{T})) : \ \ell_L(Y) = 0, \ \ell_R(Y) = 0, \ \ell_{0:N}(Y) = 0 \right\}. \tag{6}$$

This constrained optimisation problem can be solved with the iterated extended Kalman smoother (IEKS). The IEKS is a state-space implementation of a Gauss–Newton algorithm [17]. As such, one step of the IEKS computes the closed-form minimum of Equation (6) with a Kalman smoother, where the non-linear $\ell_{0:N}$ is replaced by its first-order Taylor approximation around the previous iterate. Under mild assumptions on the non-linearity of $f$ and the magnitude of the objective at the optimum, Gauss–Newton methods are locally convergent with linear rate [18].

Each iteration of the IEKS returns a mean and covariance function. Eventually, the scheme approaches a variant of the Laplace approximation of the posterior (note the shorthand of Equation (5))

$$Y_{\text{MAP}}(t) \sim \mathcal{N}(m_{\text{MAP}}(t), C_{\text{MAP}}(t)) \approx p(Y(t) \,|\, \ell_L, \ell_{0:N}, \ell_R), \tag{7}$$

(this is a non-standard Laplace approximation in so far as it employs a Gauss–Newton approximation of the Hessian). A more detailed explanation is in Appendix A. The mean $m_{\text{MAP}}(t)$ is the MAP estimate, because it minimises the objective in Equation (6). The covariance $C_{\text{MAP}}(t)$ is the inverse (approximate) Hessian of the negative log-posterior distribution, evaluated at $m_{\text{MAP}}(t)$.

The Gaussian posterior returned by the IEKS is a probabilistic BVP solution. Thus, this basic version of the algorithm is already a valid BVP solver. But some degrees of freedom remain, whose efficient selection improves performance significantly. These will be the concern of the remainder of this work. Table 1 presents an outline.

Table 1: Configuration of the remaining degrees of freedom.

| What? | How? | Where? |
|---|---|---|
| Initialisation of the IEKS | ODE filter with Gaussian bridge | Section 3 |
| Mesh $\mathbb{T}$ | Error control | Section 4 |
| Diffusion $\sigma$ | Quasi-maximum likelihood estimation | Section 5 |
| Initial parameters $m_0, C_0$ | Expectation-maximisation | Section 5 |

## 3 An initial guess is not strictly necessary

Like every optimisation algorithm, the IEKS needs appropriate initialisation. Not only does the number of iterations depend on the proximity of the initial guess to the optimum, but BVPs often allow multiple solutions, and the algorithm can find only one of them [19, p. 10]. Non-probabilistic solvers outsource this issue to the user by expecting that an initial guess is provided.[1] While the same strategy is available for the probabilistic solver, there are natural alternatives in non-iterative Gaussian smoothers (Section 3.1), which further benefit from combination with a bridge prior (Section 3.2).

---

[1]For example, at the time of this writing, the BVP solvers in SciPy, Matlab, and DifferentialEquation.jl require the user to pass a vector of initial guesses of the solution at an initial grid to the algorithm [20–22].

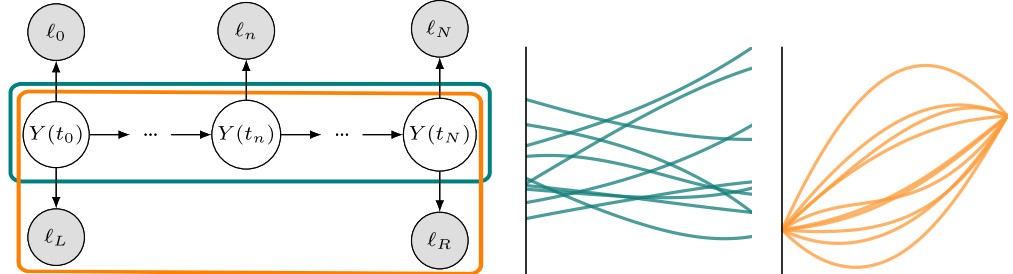

Figure 2: *Construct a bridge by considering boundary values first.* Graphical depiction of the inference problem (LEFT). Samples from the Gauss–Markov (CENTRE) and bridge prior (RIGHT).

## 3.1 Initialisation with an extended Kalman smoother

If the target of a Laplace approximation of the BVP posterior is relaxed to only *some* Gaussian approximation, an initial guess can be computed with an extended Kalman smoother (EKS) [13, 5]. Like the IEKS, the EKS linearises the non-linear ODE measurements $\ell_{0:N}$ with a first-order Taylor series. It differs from the IEKS in the position around which the approximation is constructed. The IEKS linearises all $\ell_{0:N}$ at once after each completed forwards-backwards pass. The EKS linearises each $\ell_n$ on the fly during the forward pass, at the respective predictive mean [23]. In other words, the EKS does not need an initial guess, which is why it is the tool of choice to construct one [5].

If the BVP is linear, the EKS computes the true posterior [23, 24]. If the BVP is non-linear, the EKS introduces a significant linearisation error wherever the predictive distribution deviates strongly from the true posterior. Unfortunately, in its standard implementation, the EKS necessarily starts with incomplete information about the state $y(t_0)$ and higher-order derivatives (initialisation of which is crucial to probabilistic initial value problem solvers as well [7]). Ensuring that the prior distribution satisfies the boundary conditions by construction solves this problem because the iteration can never drift too far away from the optimum. The following Section 3.2 explains more.

## 3.2 Changing the order of updates to build a bridge prior

Recall that there are three sources of information: the left-hand side boundary condition $\ell_L$, the right-hand side boundary condition $\ell_R$, and the ODE measurements $\ell_{0:N}$. If the initial and terminal state of the prior distribution are forced to accommodate $\ell_L$ and $\ell_R$ before conditioning on $\ell_{0:N}$, samples from the resulting Gaussian bridge satisfy the boundary conditions by construction; see Figure 2. The linear-time complexity of Gaussian filtering/smoothing is preserved through this change in the order of updates, because the Markov property of $Y$ yields

$$p(Y(\mathbb{T}) \,|\, \ell_L, \ell_R) = p(Y(t_0) \,|\, \ell_L, \ell_R) \prod_{n=0}^{N-1} p(Y(t_{n+1}) \,|\, Y(t_n), \ell_R). \tag{8}$$

The transition densities $p(Y(t_{n+1}) \,|\, Y(t_n), \ell_R)$ as well as the initialisation $p(Y(t_0) \,|\, \ell_L, \ell_R)$ are available in closed form (Appendix B). A reader familiar with the prediction-correction nature of Gaussian filtering can think of the implementation as follows: Roughly speaking, each prediction step of the EKS with a bridge prior involves extrapolating from the current state to the terminal state, conditioning on the boundary condition $\ell_R$, and smoothing back to the current state. Therefore, the computational complexity of an EKS forward-backward pass with the bridge prior is about twice as large compared to an EKS forward-backward pass with the conventional prior. Precise derivations are in Appendix B.

Figure 3 shows that this extra cost is made up for by the improved linearisation behaviour because encoding the boundary conditions into the prior improves the initialisation drastically. Following the forwards-backwards pass with the EKS, the IEKS requires only a few more iterations to find a fixed point similar to the truth. Not using either the bridge prior or the EKS results in an initial guess that takes more iterations to find a fixed point of a lower approximation quality. Abandoning both options, which aligns with initialisation of traditional BVP solvers, is least efficient since it converges to an inaccurate fixed-point.

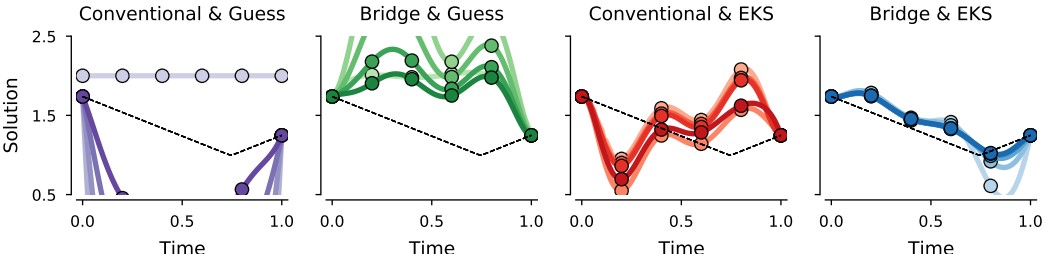

Figure 3: *In combination, EKS and bridge prior initialise well.* Initialisation and five iterations of the IEKS depicted from light to dark on the 20th problem in [25] (truth in black). Without bridges, and with an initial guess of constant twos, the fixed-point of the Gauss–Newton scheme is inaccurate on $N = 6$ points (LEFT). Using either a bridge prior (CENTRE LEFT) or the EKS (CENTRE RIGHT) lessens this problem. The bridge/EKS combination finds an accurate estimate almost immediately (RIGHT), because due to the bridge prior, the EKS linearises around a more accurate location than it would with a conventional prior during the first forward-pass.

Linear BVPs undo the effect of the bridge prior because the full posterior is computed accurately with a conventional Kalman smoother [23]. Likewise, an IEKS iteration linearises all $\ell_{0:N}$ at once, outside of the forwards-backwards pass, which renders the bridge obsolete as well. In other words, the changed order of updates is only relevant for the initialisation.

Of course, a fixed point of the IEKS is not necessarily a reliable BVP solution: its accuracy depends on the number and distribution of mesh points. The following Section 4 develops a principled and probabilistic approach to error control and mesh refinement in the BVP solver.

## 4    Estimate the error and refine the mesh

So far, the mesh $\mathbb{T}$ was assumed as given. The larger the size of this mesh is, the more accurate the solution becomes; but computational cost grows linearly with the mesh size. Low error tolerances thus require smart meshing via error control. There are two (plus one) natural candidates for error estimators, all of which connect to the probabilistic formulation of solving BVPs.

**Standard deviation:** The output of the IEKS is a Gaussian process, which can be evaluated at any point in the domain of the boundary value problem [24, Chapter 10]. Its associated standard deviation provides an error estimator. The advantage over the alternatives explained below is that it comes (essentially) for free as part of the dense output of the posterior. A potential downside of this intrinsic error estimator is its dependence on the calibration of a hyperparameter (more on this in Section 5).

**Residual:** The inference problem (Equation (5) and Figure 2) is constructed by conditioning the prior $Y$ on attaining consistently small values in its residual $\ell(Y)(t) = Y_1(t) - f(Y_0(t), t)$. Recall that if $Y_0$ were the true ODE solution, and $Y_1$ were its derivative, $\ell(Y)$ would be zero on the whole domain. Thus, the residual of the posterior mean of the approximate ODE solution estimates the error, which is a common approach in traditional, non-probabilistic algorithms as well (for instance [19] or [1, Section 9.5.1]). On a side note, considering the full posterior *distribution* implies that the residual would be a deterministic transformation of a random variable. Thus – in principle – a random variable might make a more appropriate model for the residual error than a point estimate (see Remark 1). However, this quantity will reveal itself as inaccurate in the benchmarks below.

**Remark 1.** *For a Gaussian process posterior $Y$, the law of $\ell(Y)$ is intractable in general. Linearisation of $\ell$ (at the previous iterate, like in the IEKS) unlocks a Gaussian approximation: denote the Gaussian random variable $Z(t) \approx \ell(Y)(t)$. An upper bound of the probability of $\|Z\|$ exceeding some tolerance,*

$$p\left(\|Z(t)\|^2 > tol^2\right) < \left(Trace[Cov(Z(t))] + \|\mathbb{E}(Z)(t)\|_2^2\right) / tol^2, \qquad (9)$$

*is due to the Markov inequality and a third approach to error control. The numerator of the right-hand side will be treated as an error estimator in the benchmarks below. The main difference to the point estimate is that the probabilistic version punishes magnitude* and uncertainty *in the residual.*

All three options (which we denote by a generic $e$ from now on) estimate the error at a given $t$. For mesh refinement, however, it is more instructive to consider the accumulated error on each interval

$$\epsilon_n := \left( \int_{t_n}^{t_{n+1}} \|e(t)\|_2^2 \, \mathrm{d}t \right)^{1/2}, \quad n = 0, ..., N-1. \tag{10}$$

If each $\epsilon_n$ is sufficiently small, the BVP solution is adequately accurate and the mesh appropriately fine. On those intervals where $\epsilon_n$ is too large, we introduce new grid points as follows. Assuming that the integrated error is of order $\rho > 0$, $\epsilon_n \in \mathcal{O}(h^\rho)$, splitting the interval into two equally large parts reduces the error by a factor $2^{-\rho}$, and splitting it into three equal parts by a factor $3^{-\rho}$. We use these threshold values to guide where to introduce one point and where to introduce two points. Like Kierzenka and Shampine [26], we never introduce more than two at once. For the experiments herein, and $\nu$-times integrated Wiener processes, we use $\rho = \nu + 1/2$ (which has not been proved yet but seems like a reasonable conjecture in light of Theorem 3 of Tronarp et al. [5] and our experiments).

The integral that underlies $\epsilon_n$ can usually not be computed in closed form but needs to be approximated by a numerical integration scheme. We use Bayesian quadrature (BQ) [27]. Not only does it fit neatly into the probabilistic framework, but it also allows us to place quadrature nodes freely in each domain $[t_n, t_{n+1})$. If viewed as an integral from 0 to 1, we choose quadrature nodes at $0, 0.33, 0.5, 0.67, 1$. These locations include the boundary points of the domains (0 and 1), as well as the nodes that will be introduced in case the error is too large (either 0.5, or 0.33 and 0.67). This has the advantage that, at the start of the next iteration, we reuse the evaluation of the posterior at the new mesh points. If the residual estimates the error, there is another advantage. Since the IEKS approximation is a minimum of a constrained optimisation task, the residual is zero at the boundaries of each interval. In this case, the integral is only computed on the three interior nodes. For the same reasons, non-probabilistic solvers with residual control usually employ Gauss–Lobatto schemes [19].

A final motivation for BQ is that we can tailor an integration kernel to $e$. For instance the following reproducing kernel Hilbert spaces (RKHSs) are known [5]: (i) the RKHS of $\nu$-times integrated Wiener process priors $Y(\cdot)$ is the Sobolev space of $(\nu + 1)$-times weakly differentiable functions; (ii) under some regularity assumptions on the ODE vector field, as well as on the (assumed to be) unique solution of the ODE, the RKHS of the residual $\ell(Y)(\cdot)$ is the Sobolev space of $\nu$-times weakly differentiable functions. Therefore, we base the BQ scheme on a $(\nu - 1/2)$th order Matérn prior, which has the same native space as the residual [28] (we use an exponentiated quadratic kernel for $\nu > 3$ because the required kernel embeddings are easier to compute [29, Appendix J]). The accuracy of the quadrature approximation improves with increasing $\nu$ [29]. Matching the BQ kernel to the ODE prior parallels the choices of quadrature schemes in non-probabilistic solvers [19], and ensures that the accuracy of the numerical integration does not limit the validity of the error estimate.

Which one is the most reliable error estimate? As a first testbed, we use the seventh in a collection of test problems for BVP solvers by Mazzia [25] (which will feature heavily in the remainder of this work). The derivative of the solution of this linear BVP approaches a singularity if a specific parameter is chosen sufficiently small (we use $10^{-3}$). This poses challenges for error estimators and mesh-refinement strategies. The error estimates are visualised in Figure 4. They suggest that at high tolerances, the standard deviation is more accurate than the residual; at low tolerances, the situation is reversed. This trend is preserved when moving to more challenging setups (see Section 6).

To conclude, the probabilistic framework introduces three options for error estimation and comes with a natural algorithm to compute accumulated errors in BQ. With everything explained so far, we can solve BVPs with an algorithm that adaptively refines the mesh when the solution is not sufficiently accurate. After each mesh refinement, the iteration is restarted. While it may be clear that the initial guess for the new IEKS implementation should be the approximate posterior from the previous computation, beginning a new Gauss–Newton scheme offers the chance to update the choice of other hyperparameters and thus set up a more appropriate probabilistic model for free (Section 5).

## 5  Calibration of hyperparameters with maximum-likelihood and EM

Thus far, an approximate BVP solution has been computed with $\sigma$, $m_0$, and $C_0$ set to default values. Maximum-likelihood estimates of these hyperparameters can be computed by coordinate ascent,

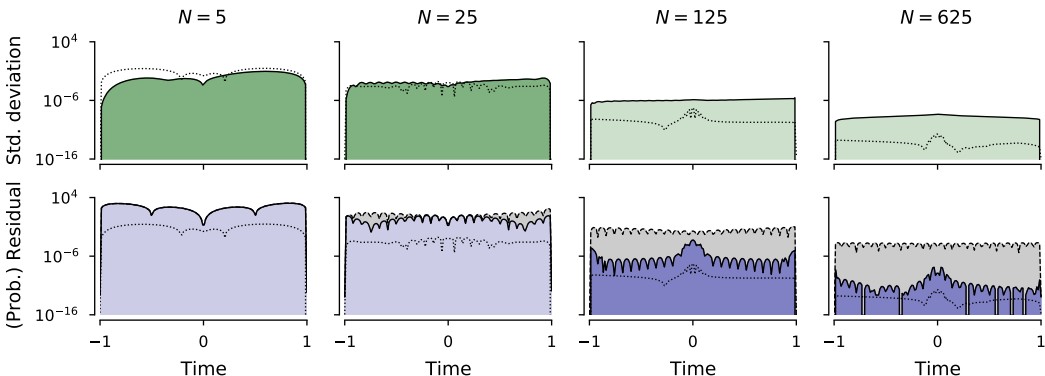

Figure 4: *Error estimation on the seventh testproblem in [25].* Evaluated at $N = 5$ (LEFT), $N = 25$ (CENTRE LEFT), $N = 125$ (CENTRE RIGHT), and $N = 625$ equidistant grid points (RIGHT). Standard deviation (TOP ROW) and residual (BOTTOM ROW) respectively the probabilistic residual (BOTTOM ROW). True error in black. A good estimate accurately measures the magnitude of the error as well as the location of large deviation. On few points, the latter is less important so the well-calibrated standard deviation provides a good estimate. On many points, it is underconfident. Since for large $N$, the location of the error becomes increasingly important, the residual should be used; the probabilistic residual is consistently underconfident. The "winners" of each column have a darker colour.

which repeats alternating updates

$$\sigma^{\text{new}} := \arg\max_\sigma \log p(\ell_L, \ell_{0:N}, \ell_R \,|\, \sigma, m_0^{\text{new}}, C_0^{\text{new}}), \tag{11a}$$

$$m_0^{\text{new}}, C_0^{\text{new}} := \arg\max_{m_0, C_0} \log p(\ell_L, \ell_{0:N}, \ell_R \,|\, \sigma^{\text{new}}, m_0, C_0), \tag{11b}$$

until some stopping criterion is satisfied [30]. A quasi-maximum likelihood update for $\sigma^{\text{new}}$ (Equation 11a) is available in closed form as a by-product of the forward-pass of each IEKS iteration. This is also true for the specific order of updates detailed previously in Section 3.2 (Proposition 2 below).

**Proposition 2.** *Assume that the initial covariance and the diffusion of the Wiener process depend multiplicatively on the scalar $\sigma^2$ (recall Equation (2)). If $\ell_L$, $\ell_R$, and $\ell_{0:N}$ are noise-free (which herein they always are), the covariance of the posterior process depends multiplicatively on $\sigma^2$ and a quasi-maximum likelihood estimate for $\sigma$ is available in closed form.*

The proof of this proposition is similar to the proof of Proposition 4 of Tronarp et al. [13] yet requires a few additional manipulations because of the boundary value information contained in the bridge. A derivation – and the precise formula for the quasi-MLE – are in Appendix C.

While $\sigma$ is tuned with quasi-maximum likelihood estimation, the parameters $m_0$ and $C_0$ of the initial distribution are separately calibrated with a single step of the expectation-maximisation (EM) algorithm [31, 32] whenever the mesh needs to be refined, which implies a restart of the IEKS. In other words, this "outer loop" around calls to the IEKS is *already* part of the computational budget; therefore, sensible updates to the initial distribution parameters $m_0$ and $C_0$ are free. The general idea of EM is to maximise a lower bound of Equation (11b) instead of maximising it directly, by computing alternating $E$- and $M$-steps. For parameter estimates in state-space models, the $E$-step of the EM algorithm is the posterior distribution in Equation (5) (see e.g. [33]), a Gaussian approximation of which is available through the IEKS: recall $Y_{\text{MAP}}(t) \sim \mathcal{N}(m_{\text{MAP}}(t), C_{\text{MAP}}(t))$. The $M$-step consists of [23, Theorem 12.5 and Algorithm 12.7]

$$m_0^{\text{new}} = m_{\text{MAP}}(t_0) \tag{12a}$$

$$\sigma^2 C_0^{\text{new}} = \sigma^2 C_{\text{MAP}}(t_0) + (m_0^{\text{new}} - m_0^{\text{old}})(m_0^{\text{new}} - m_0^{\text{old}})^\top. \tag{12b}$$

EM steps always increase the likelihood, and for exponential families, convergence to a stationary point of the likelihood function is guaranteed [34, 32]. Thus, computing alternating $E$- and $M$-steps until convergence (which we do not do) would eventually yield a good estimate of the parameters. But already in the pre-asymptotic regime and for a fixed total number of IEKS iterations, making an EM update every few steps helps convergence of the IEKS in subsequent iterations (Figure 5).

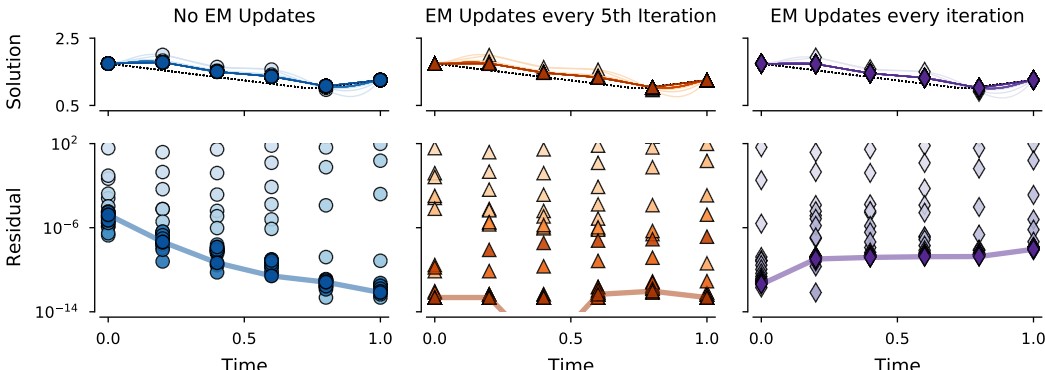

Figure 5: *EM helps the IEKS overcome unknown initial conditions.* Depicted are a fixed total of the first 25 IEKS iterations (light to dark in each respective colour) on $N = 6$ grid points, initialised with an EKS using a 7-times integrated Wiener process bridge prior on the 20th test problem in [25]. Without any EM updates to the initial condition, the convergence of the IEKS is inhibited (LEFT). EM updates every fifth IEKS iteration lead to the residual converging to zero reliably (CENTRE). Too many EM updates are not optimal either (RIGHT).

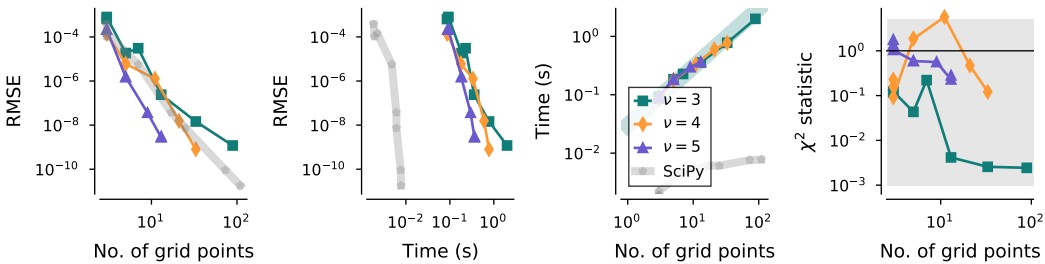

Figure 6: *Results on Bratu's problem.* The higher-order solvers converge at least as fast as the SciPy reference (LEFT) and the wall-clock time grows linearly with the number of grid points (CENTRE RIGHT; linear complexity reference line in the background). Our research code is slower by factor $\sim$ 100 than this highly optimised code base, which is a success compared to prior implementations of probabilistic BVP solvers (CENTRE LEFT). The $\chi^2$-statistic remains within $95\%$ confidence (RIGHT; intervals shaded in gray, mean $(= 1)$ in black). To show mesh refinement, the initial grid consisted of only three points; the probabilistic solver initialises with EKS and bridge, and uses the standard deviation as an error estimate.

## 6 The solver converges quickly on test problems

Now that all parts are in place, we evaluate the performance of the solver on a range of scenarios. All experiments use the CPU of a consumer-level laptop. An efficient probabilistic numerical method should provide both a good point estimate (through its posterior mean) and error estimate (through its posterior covariance). First, the approximation error should decrease rapidly with the number of grid points; we report root-mean-square errors – the lower, the better. Second, the width of the posterior distribution should be representative of the numerical approximation error (which has, to some extent, been shown in Section 4 already); we use the $\chi^2$-statistic [35]. If it is close to 1, the posterior uncertainty is calibrated. A simulation of Bratu's problem [36] for varying tolerances and orders $\nu$ suggests that the solver performs well in both metrics (Figure 6). Reassuringly, higher orders of the solver lead to faster convergence, which motivates the analysis of convergence rates akin to the analysis of Tronarp et al. [5] for initial value problem solvers. The experiments also suggest that the uncertainties are calibrated but tend to be under-confident. This phenomenon is known from probabilistic initial value problem solvers [cf. 6]. It has been studied for general Gaussian process approximations by [37].

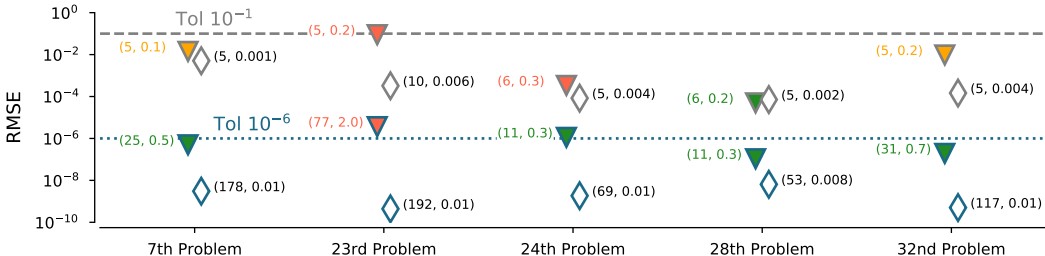

Figure 7: *The solver efficiently computes (mostly) calibrated posteriors on many problems.* Probabilistic solver ($\triangledown$; $\nu = 6$) versus SciPy's BVP solver ($\diamondsuit$). Markers are annotated with the number of grid points and runtime (in seconds). The tolerances are $10^{-1}$ (GRAY) and $10^{-6}$ (BLUE). The closer a coloured marker is to its reference line, the better. The fewer grid points and the less time required the better. Fill-color describes calibration: $\chi^2$ is within 80 % (GREEN), within 99% (ORANGE), or outside of these ranges (RED). SciPy does not allow a notion of calibration.

Efficient mesh refinement and fast convergence are evident when considering a wider range of test problems. Figure 7 depicts the results of simulating five BVPs (all from Mazzia [25]): the 7th problem approaches a singularity in its derivative, the 23rd problem has a boundary layer at $t_{\max}$, the 24th problem describes a fluid mechanical model of a shock wave, the 28th problem has a corner layer at $t_{\min}$, and the 32nd problem involves fourth-order derivatives. On all problems, the probabilistic solver efficiently computes calibrated posteriors at specified tolerances.

## 7   Related work

How does the proposed algorithm fit into the context of state-of-the-art probabilistic and non-probabilistic BVP solvers? Headway on the probabilistic solution of BVPs has been made by Hennig and Hauberg [10], Arvanitidis et al. [11], and John et al. [12]. Hennig and Hauberg [10] and Arvanitidis et al. [11] focus on the application of BVP solvers to Riemannian statistics. None of the three algorithms exploit the state space structure of the prior with its beneficial computational complexity, nor are they concerned with error estimation, mesh refinement, and the other computational aspects to the extent that this work is. Other, conceptually different probabilistic algorithms for BVPs have been proposed by Skilling [38], Chkrebtii et al. [39], Conrad et al. [40], O'Leary and Harker [41].

In terms of accuracy and cost, the present approach should rather be compared to off-the-shelf non-probabilistic BVP solvers: for instance, those implemented in Matlab [42, 19, 26], Python/SciPy [43], and Julia [44]. These toolboxes contain algorithms that implement collocation formulas and gain linear-time complexity from sparse system matrices. The Markov property makes our algorithm equally fast (in terms of the number of grid points $N$) (Table 2). The computational complexity of Algorithm 1 is $O(I_{\mathrm{Mesh}} I_{\mathrm{IEKS}} N \nu^3 d^3)$, where $I_{\mathrm{IEKS}}$ is the number of IEKS iterations, and $I_{\mathrm{Mesh}}$ is the number of mesh refinements. In our experiments, we found $I_{\mathrm{IEKS}}$ to be small, usually bounded by 10. The mesh refinement is designed to make $I_{\mathrm{Mesh}}$ as small as possible. Linear

---

**Algorithm 1:** BVP Solver

**Input:** BVP, mesh, order ($\nu$), tolerances.
**Output:** Probabilistic BVP Solution
Initialise with bridge and ODE filter;
**while** $\exists \geq 1$ *interval with large error* **do**
    Run IEKS;
    Update $m_0$ and $C_0$ (Equation (12));
    Update $\sigma$ ;
    Compute error between gridpoints;
    Refine mesh where necessary;
**end**

---

complexity in $N$ stems from the state space implementation of the IEKS and could potentially be reduced to $\log N$ by temporal parallelisation [45]. The cubic complexity in $\nu$ and in $d$ stems from the matrix-matrix operations that are required in a Kalman filter step [23]. Cubic complexity in $d$ suggests that high-order BVPs should be solved directly, without transforming them into first-order [46]. This is not uncommon for BVP solvers [1, Section 5.6] and is used herein (a version of $\ell_n$ that is suitable to high order ODEs is explained in Appendix D).

Table 2: Comparison of probabilistic and non-probabilistic BVP solvers.

|  | Non-probabilistic | Probabilistic (present work) |
| --- | --- | --- |
| $O(N)$ achieved by | Sparse matrices | Markov property |
| Error estimates | Residual (point estimate) | Many options, e.g. standard deviation |
| Initial guess | Mandatory | Optional |
| Uncertainty quantification | No | Yes |

## 8 Conclusion

We have arguably provided the first *practically usable* probabilistic BVP solver. Our method achieves the same linear computational complexity as off-the-shelf solvers, with high-quality point estimates and calibrated uncertainty. Algorithmic parameters can be set automatically by the method, including some that have to be set manually for non-probabilistic solvers. Our method thus closes a methodological gap in the toolbox of probabilistic numerics.

## Acknowledgements

The authors gratefully acknowledge financial support by the German Federal Ministry of Education and Research (BMBF) through Project ADIMEM (FKZ 01IS18052B). They also gratefully acknowledge financial support by the European Research Council through ERC StG Action 757275 / PANAMA; the DFG Cluster of Excellence "Machine Learning - New Perspectives for Science", EXC 2064/1, project number 390727645; the German Federal Ministry of Education and Research (BMBF) through the Tübingen AI Center (FKZ: 01IS18039A); and funds from the Ministry of Science, Research and Arts of the State of Baden-Württemberg. Moreover, the authors thank the International Max Planck Research School for Intelligent Systems (IMPRS-IS) for supporting Nicholas Krämer.

The authors thank Nathanael Bosch, Filip Tronarp, and Jonathan Schmidt for valuable discussions. They are grateful to Georgios Arvanitidis for helping with the implementation of the manifold example in Figure 1.

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
