# Appendix: Linear-Time Probabilistic Solutions of Boundary Value Problems

**Nicholas Krämer**
University of Tübingen
Tübingen, Germany
nicholas.kraemer@uni-tuebingen.de

**Philipp Hennig**
University of Tübingen and
Max Planck Institute for Intelligent Systems
Tübingen, Germany
philipp.hennig@uni-tuebingen.de

## A  IEKS as a Gauss–Newton Laplace approximation

The present appendix shows that the iterated extended Kalman filter (IEKS) yields a Gauss–Newton Laplace approximation of the posterior distribution (recall Equation (6) from the main paper)

$$p(Y(\mathbb{T}) \mid \ell_L(Y) = 0, \ \ell_R(Y) = 0, \ \ell_{0:N}(Y) = 0). \tag{A.1}$$

Let $\delta$ be the Dirac delta. Introduce the random variables $Z_L$, $Z_R$, $Z_{0:N}$ as

$$Z_L \mid Y(t_0) \sim \delta(\ell_L(Y)), \quad Z_R \mid Y(t_{\max}) \sim \delta(\ell_R(Y)), \quad Z_n \mid Y(t_n) \sim \delta(\ell_n(Y)(t_n)). \tag{A.2}$$

The posterior distribution in Equation (A.1) becomes

$$p(Y(\mathbb{T}) \mid Z_L = 0, \ Z_R = 0, \ Z_{0:N} = 0). \tag{A.3}$$

The difference between Equation (A.1) and Equation (A.3) is only notational. The reformulation in terms of $Z$ will be useful in the next step. The prior distribution is Gaussian (recall Section 2.1),

$$p(Y(\mathbb{T})) = \mathcal{N}(m(\mathbb{T}), K(\mathbb{T}, \mathbb{T})) \tag{A.4}$$

for mean and covariance functions $m$ and $K$ that correspond directly to the stochastic differential equation (SDE) representation in Equation 2 in the main paper [1, Chapter 12]. The use of this general Gaussian process formulation, as opposed to a sequential notion that exploits the Markov property, will simplify the notation (and simultaneously slightly generalise the result).

To show that the IEKS provides a Gauss–Newton version of the Laplace approximation, it is instructive to consider a relaxed version of the Dirac likelihoods; that is, let $\lambda > 0$ and (re)define (recall the ODE $\dot{y} = f(y, t)$, and the boundary conditions $Ly(t_0) = y_0$, $Ry(t_{\max}) = y_{\max}$),

$$Z_L \mid Y(t_0) \sim \mathcal{N}(LY_0(t_0) - y_0, \lambda I), \tag{A.5a}$$

$$Z_R \mid Y(t_{\max}) \sim \mathcal{N}(RY_0(t_{\max}) - y_{\max}, \lambda I), \tag{A.5b}$$

$$Z_n \mid Y(t_n) \sim \mathcal{N}(Y_1(t_n) - f(Y_0(t_n), t_n), \lambda I). \tag{A.5c}$$

In these formulas, $I$ is always an identity matrix of appropriate size. The limit $\lambda \to 0$ recovers the Dirac likelihoods used in the previous paragraph and in the paper. In the Gaussian relaxation, the MAP estimate is the argument that minimises the following objective (the negative log-posterior),

$$\arg\min_{Y(\mathbb{T})} \frac{1}{2}\mathcal{V}_1(Y(\mathbb{T})) + \frac{1}{2}\mathcal{V}_2(Y(\mathbb{T})) + \frac{1}{2}\mathcal{V}_3(Y(\mathbb{T})), \tag{A.6}$$

which uses the abbreviations

$$\mathcal{V}_1(Y(\mathbb{T})) := \|m(\mathbb{T}) - Y(\mathbb{T})\|^2_{K(\mathbb{T},\mathbb{T})^{-1}}, \tag{A.7a}$$

$$\mathcal{V}_2(Y(\mathbb{T})) := \frac{1}{\lambda}\|LY_0(t_0) - y_0\|^2 + \frac{1}{\lambda}\|RY_0(t_{\max}) - y_{\max})\|^2, \tag{A.7b}$$

$$\mathcal{V}_3(Y(\mathbb{T})) := \frac{1}{\lambda}\sum_{n=0}^{N}\|Y_1(t_n) - f(Y_0(t_n), t_n)\|^2. \tag{A.7c}$$

35th Conference on Neural Information Processing Systems (NeurIPS 2021).

$\mathcal{V}_1$ captures the prior distribution, $\mathcal{V}_2$ the boundary conditions, and $\mathcal{V}_3$ the (artificial) ODE "measurements". Only $\mathcal{V}_3$ includes non-linearities.

Let $\xi = (\xi_0, ..., \xi_N) \in \mathbb{R}^{(N+1) \times d(\nu+1)}$ (there are $N + 1$ points in $\mathbb{T}$) be the result of a previous Gauss–Newton iteration (or the initialisation, respectively). Denote by $P_0$ and $P_1$ the projection matrices from $Y$ to $Y_0$, and $Y$ to $Y_1$, respectively. Gauss–Newton optimisers such as the IEKS iteratively linearise the non-linearities of the objective "inside the norm" at $\xi$, and solve the resulting linear least-squares problem in closed form [2]. In other words, let

$$f(y, t_n) \approx f(P_0\xi_n, t_n) + \nabla f(P_0\xi_n, t_n)(y - P_0\xi_n) \tag{A.8}$$

be the first-order Taylor series linearisation of $f$ at $P_0\xi_n$ (each grid-point $t_n$ uses a different $\xi_n$). Then, the IEKS minimises

$$\mathcal{V}(Y(\mathbb{T})) := \frac{1}{2}\|m(\mathbb{T}) - Y(\mathbb{T})\|^2_{K(\mathbb{T}, \mathbb{T})^{-1}} + \frac{1}{2\lambda}\|FY(\mathbb{T}) + b\|^2, \tag{A.9}$$

which uses the batch notation (abbreviate $F_n := P_1 - \nabla f(P_0\xi_n, t_n)P_0$),

$$F = \begin{pmatrix} L & 0 & \dots & 0 \\ 0 & \dots & \dots & R \\ F_0 & 0 & \dots & 0 \\ 0 & F_1 & \ddots & \vdots \\ \vdots & \ddots & \ddots & \vdots \\ 0 & \dots & \dots & F_N \end{pmatrix}, \quad b = \begin{pmatrix} y_0 \\ y_{\max} \\ \nabla f(\xi_0, t_0)\xi_0 - f(\xi_0, t_0) \\ \nabla f(\xi_1, t_1)\xi_1 - f(\xi_1, t_1) \\ \vdots \\ \nabla f(\xi_N, t_N)\xi_N - f(\xi_N, t_N) \end{pmatrix}. \tag{A.10}$$

This is a linear least-squares problem and can be solved in closed form with GP regression – or, as in the present setting, with a Kalman smoother [3]. The mean of this solution becomes the new iterate $\xi$. Unless a fixed point has been found, the procedure is repeated.

The Hessian of the objective in Equation (A.9)[1] and its inverse are

$$\nabla^2 \mathcal{V}(x) := K(\mathbb{T}, \mathbb{T})^{-1} + \frac{1}{\lambda}F^\top F, \tag{A.11a}$$

$$(\nabla^2 \mathcal{V}(x))^{-1} := K(\mathbb{T}, \mathbb{T}) - K(\mathbb{T}, \mathbb{T})F^\top(FK(\mathbb{T}, \mathbb{T})F^\top + \lambda)^{-1}FK(\mathbb{T}, \mathbb{T}). \tag{A.11b}$$

The functional form of the inverse is revealed by, for instance, the matrix inversion lemma [5]. The inverse Hessian identifies as the posterior covariance of Gaussian process regression (respectively the Kalman smoother) [6, 1].

At the final iteration of the IEKS, the objective is linearised at the MAP estimate ($\xi = m_{\mathrm{MAP}}$). This fixed point then yields a Gaussian approximation of the posterior, where the mean is the MAP estimate, and the covariance is the negative inverse Gauss–Newton Hessian of the log-posterior, evaluated at the MAP estimate. This shows how the IEKS yields a Gauss–Newton Laplace transform of the relaxed objective. The limit $\lambda \to 0$ translates this to the Dirac objectives used in the paper.

In summary, the IEKS yields a Gauss–Newton Laplace approximation of the posterior, because (i) it uses a Gauss–Newton approximation of the non-linear objective, which (ii) can then be solved in closed form with a Kalman smoother, which – since it delivers Gaussian posteriors – is (iii) its own Laplace approximation. Put differently, each iteration of the IEKS yields a Gaussian approximation of the posterior, where the covariance is the negative inverse Hessian of the log-posterior, evaluated at the mean – when the mean converges to the MAP estimate, this makes the IEKS compute a Laplace approximation.

## B  Transition densities of the bridge prior are available in closed form

The present section describes the transition densities of the bridge prior. Recall the SDE representation of the prior process $Y$ (Equation (2) in the main paper). Due to the Markov property, the law of $Y$ factorises as

$$p(Y(\mathbb{T})) = p(Y(t_0)) \prod_{n=1}^{N} p(Y(t_n) \mid Y(t_{n-1})). \tag{B.1}$$

---

[1]We call the Hessian of the Gauss–Newton objective as the Gauss–Newton Hessian of the full objective [4].

The initial distribution

$$p(Y(t_0)) = \mathcal{N}(m_0, \sigma^2 C_0) \tag{B.2}$$

is part of the prior model (Section 2.1 in the main paper). The transition densities

$$p(Y(t_{n+1}) \mid Y(t_n)) = \mathcal{N}(\Phi(t_{n+1}, t_n) Y(t_n), \sigma^2 Q(t_{n+1}, t_n)) \tag{B.3}$$

use the definitions [1],

$$\Phi(t, s) := \exp(A(t - s)) \tag{B.4a}$$

$$Q(t, s) := \int_0^{t-s} \Phi(t, \tau) B B^\top \Phi(s, \tau)^\top \, d\tau. \tag{B.4b}$$

Both quantities can be computed with, e.g., matrix fractions [1]. $A$ and $B$ stem from the SDE representation (Equation (2) in the main paper). The process noise covariance is of the form $\sigma^2 Q(t, s)$ because the diffusion of the Wiener process is (by assumption) $\Gamma = \sigma^2 I$ (and the diffusion of the Wiener process would enter Equation (B.4b) as $BB^\top \rightsquigarrow B\Gamma B^\top$ [1]).

## B.1 Initial distribution

The first objective of the present section is the parametrisation of the updated initial distribution (recall the shorthand for the information sources, first introduced in Equation (7) in the main paper)

$$p(Y(t_0) \mid \ell_L, \ell_R). \tag{B.5}$$

It arises as follows. The joint distribution is

$$p(Y(t_0), \ell_L, \ell_R) = \mathcal{N}(\xi, \sigma^2 \Xi), \tag{B.6}$$

with

$$\xi := \begin{pmatrix} m_0 \\ L m_0 - y_0 \\ R \Phi(t_{\max}, t_0) m_0 - y_{\max} \end{pmatrix}, \quad \Xi := \begin{pmatrix} \Xi_1 & \Xi_2 & \Xi_3 \\ \Xi_2^\top & \Xi_4 & \Xi_5 \\ \Xi_3^\top & \Xi_5^\top & \Xi_6 \end{pmatrix}, \tag{B.7}$$

where we abbreviated

$$\Xi_1 := C_0, \tag{B.8a}$$

$$\Xi_2 := C_0 L^\top, \tag{B.8b}$$

$$\Xi_3 := C_0 \Phi(t_{\max}, t_0)^\top R^\top, \tag{B.8c}$$

$$\Xi_4 := L C_0 L^\top, \tag{B.8d}$$

$$\Xi_5 := L C_0 \Phi(t_{\max}, t_0)^\top R^\top, \tag{B.8e}$$

$$\Xi_6 := R \left[ \Phi(t_{\max}, t_0) C_0 \Phi(t_{\max}, t_0)^\top + Q(t_{\max}, t_0) \right] R^\top. \tag{B.8f}$$

Mean and covariance of $Y(t_0)$ conditioned on $\ell_L$ and $\ell_R$ now follow from standard conditioning rules of Gaussian distributions [6]. Since $C_0$ and the process noise of the covariance depend multiplicatively on $\sigma$, so does $\Xi$.

## B.2 Transition densities

Let $Y(t_n) \sim \mathcal{N}(m_n, \sigma^2 C_n)$ and recall $\ell_L$ and $\ell_R$. The second objective of the present section is the transition density from $Y(t_{n-1})$ to $Y(t_n)$ under acknowledgement of the boundary conditions. The joint distribution of $Y(t_{n+1})$ and the right-hand side boundary condition, given $Y(t_n)$, is

$$p(Y(t_{n+1}), \delta_R \mid Y(t_n)) = \mathcal{N}(\zeta, \sigma^2 \Lambda) \tag{B.9}$$

with

$$\zeta = \begin{pmatrix} \Phi(t_{n+1}, t_n) m_n \\ R \Phi(t_{\max}, t_n) m_n - y_{\max} \end{pmatrix}, \quad \text{and} \quad \Lambda = \begin{pmatrix} \Lambda_1 & \Lambda_2^\top \\ \Lambda_2 & \Lambda_3 \end{pmatrix}, \tag{B.10}$$

which uses the abbreviations

$$\Lambda_1 := \Phi(t_{n+1}, t_n) C_n \Phi(t_{n+1}, t_n)^\top + \sigma^2 Q(t_{n+1}, t_n) \tag{B.11a}$$

$$\Lambda_2 := \Lambda_1 \Phi(t_{\max}, t_{n+1})^\top R^\top \tag{B.11b}$$

$$\Lambda_3 := R \left[ \Phi(t_{\max}, t_{n+1}) \Lambda_1 \Phi(t_{\max}, t_{n+1})^\top + \sigma^2 Q(t_{\max}, t_{n+1}) \right] R^\top. \tag{B.11c}$$

Notably, since the covariance of $Y(t_n)$ depends multiplicatively on $\sigma^2$, all entries of $\Lambda$ do as well (this will be useful in Appendix C). Finally, the distribution $p(Y(t_{n+1}) \mid \ell_R, Y(t_n))$ is Gaussian with mean and covariance that are available with the usual conditioning formula for multivariate Gaussians [6]. On a side note: $\Lambda_3$ is ill-conditioned for $t_{\max} \approx t_{n+1}$, which is a problem that can be solved with appropriate preconditioning as well as square-root implementation [7].

## C  The quasi-MLE is essentially unaffected by the bridge

The present section proves that the quasi-maximum likelihood estimate (quasi-MLE) for the diffusion $\sigma$ is available in closed form, even for the bridge prior. A formula is given as well. We say that a matrix $X$ depends multiplicatively on $\sigma^2$, if it satisfies $X = \sigma^2 \check{X}$ for some $\check{X}$. First, we need to establish that all the covariances that contribute to the (approximate) prediction error decomposition depend multiplicatively on $\sigma^2$. This has partly been done in Appendix B. Second, this multiplicative dependency gives rise to a closed-form solution for the quasi-MLE.

Tronarp et al. [8] establish that for a conventional Gauss–Markov prior, and noise-free ODE measurements, the covariances of the predictive distribution $p(Y(t_{n+1} \mid Y(t_n))$ depend multiplicatively on $\sigma^2$. Appendix B established the same for the predictive distribution $p(Y(t_{n+1}) \mid \delta_R, Y(t_n))$ of the bridge.

The same will hold not only for the predictive distribution, but also for the filtering covariances, as shown next. The (iterated) extended Kalman filter approximates the non-linear ODE likelihood [9, 10]

$$p(\ell_n \mid Y(t_n)) = \delta(Y_1(t_n) - f(Y_0(t_n), t_n)) \tag{C.1}$$

with a first order Taylor approximation around some $\xi_n \in \mathbb{R}^{d(\nu+1)}$ (recall from Appendix B that $P_0$ is the projection matrix from $Y$ to $Y_0$),

$$p(\ell_n \mid Y(t_n)) \approx \delta(Y_1(t_n) - \nabla f(P_0 \xi_n, t_n)(Y_0(t_n) - P_0 \xi_n)). \tag{C.2}$$

For the non-iterated Kalman filter, $\xi_n$ is the mean of the predictive distribution [10]; for the iterated extended Kalman smoother, $\xi_n$ is the mean of the previous iteration [3]. Since this is a noise-free (i.e. Dirac) likelihood, the law of $\ell_n$ given $Y(t_n)$ is Gaussian with a covariance that depends multiplicatively on $\sigma^2$. Therefore, the covariance of $Y(t_{n+1})$ conditioned on $\ell_n = 0$ (approximately, with an [iterated] extended Kalman filter), depends multiplicatively on $\sigma^2$ as well.

Consider the following take on the prediction error decomposition [11],[2]

$$p(\ell_L, \ell_{0:N}, \ell_R \mid \sigma) = p(\ell_0 \mid \ell_L, \ell_R, \sigma) p(\ell_R \mid \ell_L, \sigma) p(\ell_L \mid \sigma) \prod_{n=1}^{N} p(\ell_n \mid \ell_{n-1}, \ell_R, \sigma) \tag{C.3}$$

which mirrors the factorisation of the prior in Equation (8) of the main paper. All of the terms in Equation (C.3) are (approximated by) Gaussian distributions (which has been shown above),

$$p(\ell_L \mid \sigma) = \mathcal{N}(z_L, \sigma^2 S_L) \tag{C.4a}$$

$$p(\ell_R \mid \ell_L, \sigma) = \mathcal{N}(z_R, \sigma^2 S_R) \tag{C.4b}$$

$$p(\ell_0 \mid \ell_L, \ell_R, \sigma) \approx \mathcal{N}(z_0, \sigma^2 S_0) \tag{C.4c}$$

$$p(\ell_n \mid \ell_{n-1}, \ell_R, \sigma) \approx \mathcal{N}(z_n, \sigma^2 S_n) \tag{C.4d}$$

either because they are Gaussian by construction (the boundary conditions are linear), or because the (iterated) extended Kalman filter employs a Gaussian approximation. The joint likelihood of $\ell_L, \ell_R$,

---

[2]It is not the traditional prediction error decomposition in so far as it employs the bridge prior.

and $\ell_{0:N}$ is maximised by the term that minimises the negative log-probability of all of these random variables being zero (neglecting some additive constants that do not depend on $\sigma$),

$$-2\log p(\ell_L, \ell_{0:N}, \ell_R \mid \sigma) \approx \frac{1}{\sigma^2}\Psi_0 + \log(\sigma^2)\Psi_1 + \text{const} \tag{C.5}$$

which employs the abbreviations

$$\Psi_0 := z_L^\top S_L^{-1} z_L + z_R^\top S_R^{-1} z_R + \sum_{n=0}^{N} z_n^\top S_n^{-1} z_n, \quad \Psi_1 := d_L + d_R + d(N+1). \tag{C.6}$$

Setting the derivative of the likelihood in Equation (C.5) with respect to $\sigma$ to zero, yields

$$-\frac{2}{\sigma^3}\Psi_0 + \frac{2}{\sigma}\Psi_1 = 0 \quad \Leftrightarrow \quad \sigma^2 = \frac{\Psi_0}{\Psi_1} \tag{C.7}$$

which gives a formula for the quasi-maximum likelihood estimate of the diffusion. From the first iteration of the IEKS onwards, this quasi-MLE equals the quasi-MLE from Tronarp et al. [9]; for the initialisation via the extended Kalman smoother, the bridge prior alters the linearisation over the law of $\ell_n$, and thus affects the quasi-MLE.

## D  Solve higher-order BVPs directly

The present appendix explains how BVPs based on higher-order ODEs can be solved directly without transforming them into first-order problems. A more comprehensive explanation is provided by Bosch et al. [12]. Many problems in the test set by Mazzia [13] are second-order. The 32nd problem in [13] (which features in Section 6 of the main paper) is fourth-order.

As an instructive example, consider the second order ODE

$$\ddot{y}(t) = f(\dot{y}(t), y(t), t). \tag{D.1}$$

If $y$ and $\dot{y}$ would be stacked into a new state $z := (\dot{y}, y)$, the ODE could equivalently be written as

$$\dot{z}(t) = g(z(t), t), \tag{D.2}$$

with $g(z(t)) = f(\dot{y}(t), y(t))$. Recall from Equation (4) in the main paper that such first-order ODEs give rise to the information operator

$$\ell_{\text{1st}}(Y)(t) := Y_1(t) - f(Y_0(t), t). \tag{D.3}$$

With this approach, higher-order ODEs can be solved. The increased dimensionality of the ODE problem makes this inefficient (as outlined in Section 8).

ODE information operators can straightforwardly be generalised to second-order ODEs, via

$$\ell_{\text{2nd}}(Y)(t) := Y_2 - f(Y_1(t), Y_0(t), t). \tag{D.4}$$

Higher-order ODEs, like the fourth-order ODE that has been part of the experiments, use the same concept: provided $\nu \geq 4$, we can define a likelihood for fourth-order ODEs,

$$\ell_{\text{4th}}(Y)(t) := Y_4 - f(Y_3(t), Y_2(t), Y_1(t), Y_0(t), t). \tag{D.5}$$

The only requirement for this to work is that $\nu$ is sufficiently large. All of these likelihood functions can be used inside an extended Kalman filter. Solving higher-order BVPs directly, without transforming them into first-order problems, is not uncommon for BVP solvers [14, Section 5.6].