# OpenReview forum: "Linear-Time Probabilistic Solution of Boundary Value Problems"
_NeurIPS.cc/2021/Conference — NeurIPS 2021 Poster_

### Official Review · Reviewer_iTr9 · 2021-07-02

**Rating:** 8
**Confidence:** 4

**Summary:**

This paper introduces a probabilistic boundary value problem solver that inherits much of its background from the similar  Gaussian filtering paradigm developed over several recent papers for IVP solvers. While the method initially seems like a minor extension to these, the paper discusses initialisation, calibration, mesh refinement to achieve pre-set tolerance level, things which are often neglected in these types of works. There are encouraging results (though mutliple approaches for eg. initialisation are presented and its not clear which is recommended in given cases) and the claim is made that the computational cost of the algorithm is competitive with state of the art classical solvers.

**Ethical Concerns:**

No plausible negative ethical concerns from this work.

**Limitations And Societal Impact:**

No plausible negative societal impacts from this work so not addressed.

**Main Review:**

This is a strong paper which attacks an important problem (BVPs) within the relatively new yet increasingly established probabilistic numerics paradigm. The work is rigorous and considers at length practical issues such as initialisation and computational cost (though in the latter case benchmarking is not provided, and neither is code) while also undertaking a detailed discussion of calibration, hyperparameter setting, and differences in prior choice. These issues are often neglected in similar papers. The detailed description of the model in Appendix A is informative and well written and it is a pity it does not fit into the main paper. I particularly appreciated the focus on the use of the method in practice (including what seems from the description like carefully designed code) though obviously the code has not been made available. I also appreciated the clear writing. I did wonder given the claims of 'practical usability' why no benchmarking was attempted.

I do have some questions and clarifications for the authors:
* Can this method easily extend to the situation where the boundary conditions are random (with known distribution, say, but not simply a fixed point). Would this also apply to the revised model in Sec 3.2 ?
* Fig 3. Please provide some more detail on what happens when the number of points is increased. Is the poor EKS linearisation using the guess start made up for with more points, or is it long term catastrophic? Are there any issues with the EKS scheme (beyond the additional cost), say the failure to find alternative solutions where more than one exists etc? Some comment would be good.
* It’s not discussed how to set \nu. What is the intuition with the higher derivatives? I can see in Fig 6 that you have varied this parameter but how does the practioner decide? This seems like the major critical algorithm parameter not discussed.
*Is BQ sufficiently accurate in practice with such a small number of nodes? How is the BQ uncertainty propagated to the main IEKS algorithm? That’s not entirely clear. How can you discriminate the source of error as coming from the ODE or from the BQ within the steps?
* Calibration (right hand plot of Fig 6) is interesting. This seems to be not only quite poor (underconfident as stated in the main text) but also getting worse as the number of grid points increases. Are the strategies within the current algorithm (a different approach to setting certain parameters for example) that would mitigate this? How should a practitioner who presumably wants ‘ a well calibrated method’ interpret this plot?
* Did you compare your method (in wall clock time rather than in computational complexity) to existing solvers including state of the art classical ones? Can you give anything quantitive?

**Time Spent Reviewing:**

2.5

---

> ### Author Response · Authors · 2021-08-06
> **Thank you for your review, iTr9!**
>
> Thank you so much for such a positive assessment of our work. We are glad that you appreciate the focus of the paper. In the following, we will reply to your specific questions.
>
> **Random boundary conditions.**
> Yes and no. In principle, the setup (through the probabilistic language) allows random boundary conditions (e.g. Gaussians). However, there are certain intricacies regarding being able to "propagate" uncertainty over boundary conditions through the solver into the posterior distribution (in the sense that the whole posterior reflects this uncertainty). Doing this is a difficult problem and beyond the scope of this paper.
>
> **Initialisation on a dense grid.**
> This is a great question! It is not clear to which extent the linearisation error introduced by the EKS vanishes with a large number of points. On the one hand, incorrect initial distribution parameters might lead to instability down the line (see e.g. [7]), though, on the other hand, the bridge prior ensures that the iteration does not diverge. When there is more than one solution, the EKS initialisation will find different solutions for different priors.
>
> **Choice of $\nu$.**
> Loosely speaking, $\nu$ governs the convergence speed: the more derivatives are modelled, the faster the algorithm is expected to converge (as depicted in Figure 6). So in principle, choose a large $\nu$ if you want fast convergence. However, on a formal level, such a statement would require deriving convergence rates for the BVP solver (as a function of $\nu$) which is beyond the scope of this work. (The reasons why we believe this effect of $\nu$ is true are the experiments and similar theoretical results for initial value problem solvers [5].)
>
> **Accuracy of BQ.**
> We found BQ to be sufficiently accurate. The convergence rate of BQ depends on (among other things) the choice of kernel, which we matched to the prior distribution of the BVP solver (which replicates the strategy of choosing quadrature rule in non-probabilistic solvers; see page 6 of the manuscript). The posterior uncertainty of the Bayesian quadrature approximation can be compared to the resulting error estimate to evaluate whether the quadrature or the ODE solver error are the bottleneck (which in our view, is another advantage of the probabilistic approach over non-probabilistic solvers, which have no straight-forward way of detecting such "leak"s of accuracy).
>
> **Calibration.**
> We agree with you that calibration is an interesting subject matter. Yes, the width of the posterior appears underconfident. This is likely due to the choice of quasi-MLE calibration (see also the work by Karvonen et al. (2020) who analyse this phenomenon for Bayesian quadrature). For the time being, we feel encouraged that the calibration exhibits similar properties to the calibration of initial value problem solvers (see the study in [6]) and that for instance in safety-critical applications, underconfidence is better than overconfidence.
>
> **Benchmarks.**
> Yes, we compared our method to state-of-the-art, non-probabilistic solvers in both convergence speed and wall-time (see Figures 6 and 7, both of which include a SciPy reference implementation). We agree 100% with you that such a comparison is important, especially since we claim practical usability. (To contextualise these wallclock-time results, see also our response to Reviewer 9rs3.)
>
> All in all, thank you again for your positive review! We are glad you enjoyed reading the paper, and hope that your questions were answered.
>
> References:
>
> T. Karvonen et al. Maximum likelihood estimation and uncertainty quantification for Gaussian process approximation of deterministic functions. SIAM/ASA Journal for Uncertainty Quantification. 2020.

---

> > ### Comment · Reviewer_iTr9 · 2021-08-11
> > **Response**
> >
> > Thank you for the answers to my questions. I’d have liked to see an undertaking to include (even short, passing) references to some of these issues in the final version of the paper, since the purpose is not solely for my information during the review period but because I think readers are likely to ask similar questions. Specifically the answers given to bullet points 2 and 3 - on behaviour of the linearisation and the setting of the parameter \nu - are informative, have practical consequences for potential users of the method, and would improve the paper if they were addressed in the text. In general I still think this is a strong piece of research and continue to recommend acceptance.

---

> > > ### Author Response · Authors · 2021-08-11
> > > **Response to the response**
> > >
> > > Thank you for replying, iTr9.
> > >
> > > You are right, and we will definitely address those points in the text.
> > >
> > > Thank you for continuing your recommendation of acceptance!

---

### Official Review · Reviewer_9rs3 · 2021-07-05

**Rating:** 6
**Confidence:** 4

**Summary:**

The manuscript entitled “Linear-Time Probabilistic Solution of Boundary Value Problems” introduces a Gauss–Markov prior and tailors it specifically to BVPs, which allows computing a posterior distribution over the solution in linear time, at a quality and cost comparable to that of well established, non-probabilistic methods. Existing probabilistic solvers has cubic complexity which puts severe upper limits on grid resolution. Traditional, non-probabilistic BVP solvers are very efficient but do not provide probabilistic output. Thereby, they would have to serve as black-boxes inside probabilistic programs.
The main contribution of this work is that this proposed solver can deliver uncertainty quantification, mesh refinement, and hyperparameter adaptation with linear time (although it is still much slower than the non-probabilistic BVP solvers). This results in a usable probabilistic BVP solver that is natively compatible with other parts of the statistical modelling tool-chain.



**Limitations And Societal Impact:**

The author should address more of this solver’s limitations and discuss the potential improvement in the future. Limitations can be the Markov assumption, poorer point estimation than non-probabilistic solvers, the slowness of the computation compared with the off-the-shelf non-probabilistic solvers,

**Main Review:**

In general, this paper is in a good shape. I have some comments on this paper as follows:

1.	Figure 5 shows too many EM updates are not optimal either (RIGHT). The authors should give an explanation for this phenomenon.

2.	In Figure 6, the legend sits on Subplot 3. However, this legend should belong to all these 4 subplots. Also, the legend shouldn’t cover the data points on the plots.

3.	Although the proposed solver has linear-time complexity in terms of the number of grids, it is about 100 times slower than the off-the-shelf probabilistic solver which also has linear-time complexity. Authors should comment on what causes such a slowness compared with the non-probabilistic solver.

4.	I don’t see the appendix for this paper although the paper says it has.


This paper tries to close the gap between the probabilistic and non-probabilistic BVP solvers such that the new proposed probabilistic BVP solver has a linear-time complexity which is much better than the exiting probabilistic BVP solvers, and it can deliver the algorithmic parameters automatically, including some that have to be set manually for non-probabilistic solvers. It beats non-probabilistic BVP solvers in a way that this solver is naturally compatible with other statistical modelling tool-chain. Although the experimental results show this solver is still much slower and gets poorer point estimates than the off-the-shelf non-probabilistic solvers. But it made some improvements for the probabilistic BVP solvers. This paper has its significance.

In summary, I recommend its publication in NeurIPS after those issues are addressed.

**Time Spent Reviewing:**

5 hours

---

> ### Author Response · Authors · 2021-08-06
> **Thank you for your review, 9rs3!**
>
> Many thanks for your positive review. We hope that the following clarifications address all your listed concerns and convince you to keep arguing for accepting the paper.
>
> **Too many EM updates are suboptimal.**
> You raise a great point. We agree with you that this is definitely worth a discussion. In all honesty, it is not entirely clear why too frequent EM updates are suboptimal. One possible explanation would be that a too early EM update makes the mean and covariance approximations worse (initially) rather than better. Loosely speaking, the IEKS might be thrown back into the pre-asymptotic regime and thereby struggles to find a consistently small residual. When writing the paper, we did not feel comfortable speculating in this direction, which is why we stuck with the empirical, phenomenological treatment. In any case, we are glad you brought up this point and are sorry that we cannot provide a clear answer.
>
> **Factor 100x slower runtime.**
> This is also a very interesting question, thanks a lot for asking it!
> To be blunt, we actually consider a constant factor 100 difference in runtime between our code and the highly optimised SciPy solvers to be a _great success_!
> Early probabilistic ODE solvers did not even have linear-time complexity. This only changed with the advent of filtering solvers. Even these methods, up until very recently, had a run time overhead factor on the order of 10,000 and more relative to SciPy and Matlab.
> It should be acknowledged that the reference implementation in SciPy is part of a well-optimised code base while our code (despite being crafted carefully) is not yet close to this level of code optimisation. We deliberately decided to include a runtime comparison precisely because the runtimes are now so close. This indicates that "simply" optimising the codebase may completely close the gap. To us, this shows that, after years of a community effort to get the algorithmic concepts right, probabilistic ODE solvers are now at the point where they can truly compete with non-probabilistic black-box tools. We did worry that a reviewer might misunderstand this, but we do hope that you will be able to appreciate this advancement if you compare it to previous work on probabilistic ODE solvers.
>
> Hopefully, your concerns are hereby addressed. We will do our best to incorporate these answers into a revised version of the paper (especially regarding the run time).
>
> We would like to thank you again for your thorough review and for your positive assessment of the paper, as well as for asking very interesting questions!

---

> > ### Comment · Reviewer_9rs3 · 2021-08-21
> > **Thanks for the clarifications**
> >
> > Thanks for the clarifications. Would love to see the clarifications can be shown in the paper as well. I keep my current rating.

---

> > > ### Author Response · Authors · 2021-08-25
> > > **Thanks for the follow up, 9rs3!**
> > >
> > > Thank you for the follow up, 9rs3. We are glad our clarifications were helpful.
> > >
> > > We probably misunderstand your comment (one could interpret it as a remark about not having updated the paper _yet_), but please let us be rather safe than sorry by briefly replying to that interpretation of your response.
> > >
> > > We will of course address those points in a revised version of the paper as soon as  that is possible again! At the moment, i.e. during the review process, Neurips/OpenReview does not allow revisions.
> > >
> > > We 100% agree with you that including answers to your (and other reviewer's) questions will be important.
> > > Apologies if this wasn't clear enough in our first reply.

---

### Official Review · Reviewer_zNRW · 2021-07-15

**Rating:** 6
**Confidence:** 2

**Summary:**

The paper discusses refinements to current probabilistic BVP solvers, namely the IEKS. The paper proposes new initialisation and meshing strategies for speed-up. This is an interesting paper with potentially high impact.


**Limitations And Societal Impact:**

Limitations could be clearer.

**Main Review:**

Post-discussion update. The authors have adressed my concerns, and this work does improve over earlier work substantially, and there is sufficient discussion about this. Some issues still stand, but I'm nevertheless raising my score to 6, and would be happy to see this paper accepted.

--------

The paper discusses refinements to current probabilistic BVP solvers, namely the IEKS. The paper proposes new initialisation and meshing strategies to speed-up the method. This is an interesting paper with potentially high impact.

I’m having lots of trouble understanding the papers notation and nomenclature. The paper talks of posteriors without introducing any priors or likelihoods. The math is incomplete (see below). This makes following the paper unnecessarily difficult. In NIPS papers should be accessible to wide audience, and not limited by subdomain shortcuts. The sec 3 would be easier to follow if one would mathematically define at least the gist of bridge and EKS, currently the method is only in the appendix.

The paper is otherwise well written. All the methods are sensible and the individual improvements are demonstrated also empirically.

There is strangely no empirical comparisons between the final tuned IEKS and the vanilla IEKS. I don’t understand why this is missing, since the main contribution of the paper is to improve upon IEKS (or is it..). Maybe the authors feel that this would be too trivial experiment. However, the improvements gained from the individual “tricks” have varied effects, and its not clear which one improves what part of the performance. In fig5 one already gets very low residuals with vanilla method, and in fig 6 the reference method seems way faster. The authors claim that this work is the first practically usable probabilistic solver, but the impracticality of IEKS was not shown in any way. The results should also contain ablation studies of all parts or combinations of the new ideas.

The paper proposes well-thought refinements to probabilistic BVP, but the empirical evaluation of the refinements seems incomplete or lacking.


Other comments:
* The learning problem is unclear from page 1, what is it?
* What do the ball/diamond mean in fig 1?
* The paper’s notation is a bit complex. The start of sec 2.1. does not define \nu. What is a single Y_i? The notation Y : t -> R is unnecessarily complex. \Gamma does not connect anywhere. What is “integrated” wiener (reference)? How come Y has a derivative if its Wiener?
* Why is eq 5 a posterior? It looks just a conditional distribution since we haven't placed any priors to the losses \ell. Similarly this does not look like MAP, but constrained global optimisation.
* What is p(Y(t) | ..)? Please define
* Why is \ell_n nonlinear? Eq 4 looks linear.
* eq 7 drops the zero-conditioning, how do we condition here if not 0?


**Time Spent Reviewing:**

3

---

> ### Author Response · Authors · 2021-08-06
> **Thank you for your review, zNRW!**
>
> Thank you so much for describing the paper as "interesting" and "with potentially high impact". We are glad that you enjoyed it.
>
> You raised three concerns in your review of our work: notation, a lack of priors and likelihoods,  and a lack of comparison between IEKS and the "final BVP solver". We will reply to all three points individually.
>
> **Notation.**
> We take your point regarding notation. Since this paper connects to many areas including Gauss-Markov regression, Gauss-Newton methods, Gaussian bridges (lots of "Gauss" apparently...), expectation maximisation, quasi-maximum likelihood estimation, and more, we struggled ourselves to find a compact-yet-intuitive nomenclature. We will do our best to refine the notation in an updated version of the paper.
>
> **Priors, likelihoods, posteriors.**
> In a nutshell: The posterior in Equation (5) directly corresponds to the prior in Equation (2) and the likelihoods in Equations (3-4). Nevertheless, we acknowledge that the noise-free likelihoods in Equations (3-4) together with artificial data points that are equal to zero are slightly unorthodox if viewed through the lens of "classic" Bayesian inference. In principle, the information operator is exactly the likelihood you are looking for. These technical concepts are used in probabilistic ODE solvers in general (since [13], see also [5] which uses the terminology "information operator" more prominently than [13]) to provide a technically correct formulation. This construction is common for probabilistic numerical methods in general (see the reference below).
>
>
> **IEKS comparison.**
> We understand that a comparison between the final BVP solver and a "vanilla" IEKS may seem desirable at first glance. However, it cannot be done in a meaningful way for the following reason. The IEKS is not a BVP solver, but only a single part of the BVP solver toolbox (BVP solvers are more than the optimisers they use.) The actual solver is built *around* the IEKS, by combining it with a prior distribution, initialisation & mesh refinement (both of which are orthogonal to the choice of optimiser), and hyperparameter calibration. The latter (namely, the expectation-maximisation updates) are the only concepts intertwined with the IEKS, and we hope that the evaluation in Section 5 is an ablation study regarding the impact of the EM updates on the IEKS iterations. Nevertheless, we are grateful that you brought up this question, because it gave us the chance to clarify why this very natural inquiry cannot be resolved.
>
> Some short answers to your remaining questions (please kindly do reply to this comment if we misunderstood. We would be very glad to continue the conversation and clarify any issues remaining).
>
> * "The learning problem is unclear from page 1, what is it": See our answer to the question of prior and likelihood above. Does this help?
> * "What do the ball/diamond mean in fig 1?": The red diamonds are the boundary values of the red curve, the green balls are the boundary values of the green curve. Thanks for the note, we will clarify this in the caption.
> * "The notation is a bit complex": $\nu$ is an integer that describes the number of derivatives that are modelled. The $i$th component of $Y=[Y_0, ..., Y_i, ..., Y_\nu]$ models the $i$th derivative of $Y_0$ (see also line 54). An integrated Wiener process is the time-integral of a Wiener process. As such, while a Wiener process does not admit derivatives, *integrated* Wiener processes do (see e.g. [12]).
> * "Why is eq 5 a posterior", and "What is p(Y(t) | ...)":
> see again our answer above about information operators. Did this help?  Equation (5) is the posterior according to the Gauss-Markov prior, likelihoods/measurement models as defined in Equations (3-4), and data points that are equal to zero (see also the more thorough elaboration above).
> * "Why is ell_n nonlinear": $\ell_n$ is non-linear, because it depends on the non-linear ODE vector field $f$.
> * "Eq 7 drops the zero conditioning": Equation (7) drops the zeros only for notational reasons. It is nothing but short-hand for Equation (6).
>
> We hope we were able to answer your questions. If there are further concerns that keep you from recommending acceptance, please do let us know. We would be happy to continue the discussion.
>
> All in all, thank you again for your positive assessment of our work!
>
> References:
>
> J. Cockayne et al.. Bayesian probabilistic numerical methods. SIAM Review. 2019.

---

> > ### Comment · Reviewer_zNRW · 2021-08-11
> > **discussion**
> >
> > Thanks for the explanations, they helped a lot.
> >
> > I still wonder about the comparisons. You cite three methods [10,11,12] that are existing probabilistic BVP solvers, even if they are approximative or "in quotes". For instance, the GOODE method also claims that “there exists no efficient probabilistic general-purpose solver for nonlinear BVPs” and they claim to introduce one; similar to your work. There is no discussion of GOODE in your section 7.
> >
> > If some competing methods (and one closely related) exists, one would expect to see discussions and comparisons. How come there are no comparisons? To make a claim that your method is the “first usable solver” (sec 8), one first needs to demonstrate that all other methods are not usable.

---

> > > ### Author Response · Authors · 2021-08-13
> > > **Thank you for initiating a discussion, zNRW**
> > >
> > > Thanks a lot for your reply! We are glad you found our previous explanations helpful.
> > >
> > > As you have probably noticed, GOODE is the reference number [12]. We are aware that our claims are similar to the claims in the GOODE paper, and agree with you that a discussion of the differences  between the works is important (we do explain the differences in the paper; more below).
> > >
> > > The most striking difference between GOODE and our work is that we propose a solver that computes a posterior in $O(N)$ complexity ($N$ is the number of grid points) by using priors with the Markov property. Non-probabilistic solvers like the ones implemented in SciPy also cost $O(N)$. GOODE costs $O(N^3)$, which would be prohibitively expensive for large $N$ -- thus our work closes a gap in performance between non-probabilistic and probabilistic solvers. (We also do mesh-refinement, on-the-fly calibration, and efficient initialisation -- GOODE does not.) The same points can be made when comparing to [10] and [11].
> > >
> > > We hope that lines 256f (in Section 7) as well as lines 37f (in Section 1) adequately deliver these points. In general, we would be happy to discuss further!

---

> > > > ### Comment · Reviewer_zNRW · 2021-08-16
> > > > **response**
> > > >
> > > > The relation to earlier work is now clearer, and GOODE is clearly not as scalable as your method. I'm changing my review score to 6. Thanks for the discussion!

---

### Official Review · Reviewer_j7RB · 2021-07-17

**Rating:** 6
**Confidence:** 2

**Summary:**

In this paper, a fast algorithm for the probabilistic solution of BVPs are proposed. Comparing with the classical deterministic methods, the proposed method allows computing a posterior distribution in linear time while further delivers uncertainty quantification, mesh refinement and hyperparameter adaptation.

**Limitations And Societal Impact:**

See main review.

**Main Review:**

The paper is generally written well. The mathematical derivations seem proper and the discussions on initial guess, mesh refinement and hyperparameter calibration are helpful.

My main concern about the paper is its lack of comparison with baseline methods. Bayesian approach for BVPs was discussed in many classical uncertainty quantification literatures, for example:
Chkrebtii, O. A., Campbell, D. A., Calderhead, B., & Girolami, M. A. (2016). Bayesian solution uncertainty quantification for differential equations. Bayesian Analysis, 11(4), 1239-1267.
O'Leary, P., & Harker, M. (2014, May). Inverse boundary value problems with uncertain boundary values and their relevance to inclinometer measurements. In 2014 IEEE International Instrumentation and Measurement Technology Conference (I2MTC) Proceedings (pp. 165-169). IEEE.
However, the authors didn't provide a thorough review on these previous works in the area nor situate their approach with respect to them.

Another possible pitfall is: BVP problems can be ill-posed. How does the proposed method perform on those problems?

**Time Spent Reviewing:**

5

---

> ### Author Response · Authors · 2021-08-06
> **Thank you for your review, j7RB!**
>
>
> Thank you for your positive review! We are glad that you appreciate the clarity of the paper, and that you found the sections on initialisation, mesh refinement, and hyperparameter calibration helpful. Please let us reply to your two specific concerns below.
>
> **Related work.**
> We understand your concern. Initially, we decided against positioning ourselves in relation to e.g. the paper by Chkrebtii et al., for the following reason(s). Our paper has connections to differential equation solvers (BVP solvers), Gaussian processes, state estimation, and probabilistic numerics, which makes it very difficult to cover all bases. It is a minefield to cite and relate to everything that connects to at least one of these areas.  Therefore, we opted to compare our work to only the most closely related concepts: state estimation-based probabilistic ODE solvers (e.g. [4-9, 13]), Gaussian process-based BVP solvers (e.g. [10-12]), and non-probabilistic, state-of-the-art BVP solvers (e.g. [1, 18-21, 25]). Papers like the ones you mentioned unfortunately did not fall into either category, which, as we are aware, is a hard cut-off. But we take your point and will be happy to add a reference and short discussion of the connections and differences to the mentioned papers.
>
> **Ill-posed BVPs.**
> Yes, BVPs can be ill-posed which makes numerical solution difficult - not just for probabilistic solvers. For example, there may be multiple solutions (but the algorithm can only find one of them). In this case, the output of the solver would depend crucially on the initialisation, and even our proposed initialisation strategy can also only find one of the possibilities. Other than that, we expect that the behaviour of the probabilistic algorithm mirrors the behaviour of non-probabilistic solvers in such a context.
>
> We were somewhat surprised that you gave a low score to our paper, since your review is generally positive, and all issues you raise are easy to fix.
> Hopefully, our answers above address your concerns adequately. If not, we would be very happy to discuss further on here, and would be grateful if you could clarify what keeps you from recommending acceptance.
>
> Thanks again for your feedback, we are glad you enjoyed the paper!

---

> > ### Comment · Reviewer_j7RB · 2021-08-16
> > **Thanks for the response**
> >
> > I also had concerns on the relation to earlier work. After seeing the response and the discussions with other reviewers, I'm changing my review score to 6. Thanks for the response!

---

### Decision · Program_Chairs · 2021-09-28

**Decision:**

Accept (Poster)

**Comment:**

This paper proposes and studies a probabilistic numerical method for the solution of boundary value problems (BVPs) specified by ODEs.  The contribution builds on earlier work that exploited the state-space formulation of Gaussian processes (GPs) to solve initial value problems specified by ODEs.  Considerable attention is paid to practical implementation and I believe the author(s)' claim that the method is useable out of the box (the author(s) promise that code will be released).  The NeurIPS community have historically been interested in the use of GPs to solve ODEs, and I expect that this work (which is the first practical method for BVPs) will again attract significant interest.

**Consistency Experiment:**

NeurIPS has a long history of experimentation. In 2014, NeurIPS ran an experiment in which 10% of submissions were reviewed by two independent committees to quantify the randomness in the review process. This year, we repeated a variant of this experiment to see how the quality of the review process has changed over time.  This paper was part of the experiment and was therefore assigned to two committees (consisting of reviewers, an Area Chair, and a Senior Area Chair) that reached independent decisions.  If both committees made the same recommendation, this recommendation was followed. If a single committee recommended acceptance, the paper was accepted (with the exception of a few cases in which the other committee identified what we considered a fatal flaw, e.g., an error in a key result).

This copy’s committee reached the following decision: **Accept (Poster)**

The other committee assigned to the paper recommended **Reject**.  You can find the other set of reviews, along with any follow up discussion with the authors here:
https://openreview.net/forum?id=U9NNzquYEHC